Effects of tobacco smoke and electronic cigarette vapor exposure on the oral and gut microbiota in humans: a pilot study

Stewart Christopher J. 1 2 christopher.stewart@bcm.edu
Auchtung Thomas A. 1
Ajami Nadim J. 1
Velasquez Kenia 3 4
http://orcid.org/0000-0002-2479-2044 Smith Daniel P. 1
De La Garza Richard II 3 4 5
http://orcid.org/0000-0002-1105-566X Salas Ramiro 3 4 5
Petrosino Joseph F. 1
1 Alkek Center for Metagenomics and Microbiome Research, Department of Molecular Virology and Microbiology, Baylor College of Medicine , Houston, TX , USA
2 Institute of Cellular Medicine, Newcastle University , Newcastle , UK
3 Menninger Department of Psychiatry and Behavioral Sciences, Baylor College of Medicine , Houston, TX , USA
4 Veteran Affairs Medical Center, Michael E. DeBakey VA Medical Center , Houston, TX , USA
5 Department of Neuroscience, Baylor College of Medicine , Houston, TX , USA
Houck Keith
Electronic publication date: 2018 Apr 30
Publication date: 2018
Volume: 6
Electronic Location ID: e4693
Received 2018 Mar 2; Accepted 2018 Apr 10
Copyright: © 2018 Stewart et al.
Copyright year: 2018
Copyright holder: Stewart et al.
License: This is an open access article distributed under the terms of the Creative Commons Attribution License, which permits unrestricted use, distribution, reproduction and adaptation in any medium and for any purpose provided that it is properly attributed. For attribution, the original author(s), title, publication source (PeerJ) and either DOI or URL of the article must be cited.
License URL: https://creativecommons.org/licenses/by/4.0/

Erratum in: Correction: Effects of tobacco smoke and electronic cigarette vapor exposure on the oral and gut microbiota in humans: a pilot study 6 23 8 2018 e4693/correction-1 PeerJ PMC6106372 30148001
Keywords: Smoking, Microbiota, Electronic cigarette, Tobacco

Funding: NCI 3P30CA0125123-09S1 Veteran Health Administration VHA5I01CX000994 McNair Medical Institute This work was supported by NCI (3P30CA0125123-09S1); the Veteran Health Administration (VHA5I01CX000994); and the McNair Medical Institute. This material is partly the result of work supported with resources and the use of facilities at the Michael E. DeBakey VA Medical Center, Houston, TX. The funders had no role in study design, data collection and analysis, decision to publish, or preparation of the manuscript.

==============================
Background

The use of electronic cigarettes (ECs) has increased drastically over the past five years, primarily as an alternative to smoking tobacco cigarettes. However, the adverse effects of acute and long-term use of ECs on the microbiota have not been explored. In this pilot study, we sought to determine if ECs or tobacco smoking alter the oral and gut microbiota in comparison to non-smoking controls.

Methods

We examined a human cohort consisting of 30 individuals: 10 EC users, 10 tobacco smokers, and 10 controls. We collected cross-sectional fecal, buccal swabs, and saliva samples from each participant. All samples underwent V4 16S rRNA gene sequencing.

Results

Tobacco smoking had a significant effect on the bacterial profiles in all sample types when compared to controls, and in feces and buccal swabs when compared to EC users. The most significant associations were found in the gut, with an increased relative abundance of Prevotella (P = 0.006) and decreased Bacteroides (P = 0.036) in tobacco smokers. The Shannon diversity was also significantly reduced (P = 0.009) in fecal samples collected from tobacco smokers compared to controls. No significant difference was found in the alpha diversity, beta-diversity or taxonomic relative abundances between EC users and controls.

Discussion

From a microbial ecology perspective, the current pilot data demonstrate that the use of ECs may represent a safer alternative compared to tobacco smoking. However, validation in larger cohorts and greater understanding of the short and long-term impact of EC use on microbiota composition and function is warranted.

Introduction

Tobacco cigarettes are the leading cause of preventable diseases in the world (Cahn & Siegel, 2011). Smoking increases the risk for development of several diseases, including cardiovascular disease (Lubin et al., 2016), various cancers (Jacobs et al., 2015), especially lung cancer (Montserrat-Capdevila et al., 2016), and inflammatory bowel disease (Higuchi et al., 2012). Electronic cigarettes (ECs) offer promise as a tool to quit or an alternative to tobacco smoking. It is estimated that over 12% of adults in the US have used ECs (Kosmider et al., 2016). Use of ECs is tripling annually with consumers including non-tobacco smoking adolescents and adults (Moon, Lee & Lee, 2015; Bostean, Trinidad & McCarthy, 2015). While ECs primarily contain propylene glycol, vegetable glycerin, and nicotine, tobacco cigarettes are composed of over 4,000 other chemicals and particulate matter (You et al., 2015). Studies reporting negative health effects relating to ECs are scarce and ECs remain unregulated, but commercial ECs have been reported to contain low levels of toxic compounds (Cahn & Siegel, 2011; Varlet et al., 2015; Kosmider et al., 2016; Allen et al., 2016).

There are relatively few studies exploring the effects of tobacco smoke on the microbiota and we are not aware of any study to date that has compared the bacterial communities in tobacco smokers and EC users. In one human study, the oral microbiota was altered between healthy non-smokers and tobacco smokers, with decreased Porphyromonas, Neisseria, and Gemella in tobacco smokers, but the lung communities were not affected (Morris et al., 2013). Smoking has also been shown to drive changes in the sputum microbiota more than other lifestyle factors (e.g., exercise and alcohol), increasing the relative abundance of Veillonella and Megasphaera (Lim et al., 2016). A recent large-scale sequencing study of the oral microbiota in current, previous, or non-smokers demonstrated current smokers had distinct oral communities, with reduced relative abundance of Proteobacteria (Wu et al., 2016). Notably, the significant taxa vary between studies and a recent analysis of numerous sites within the mouth found no significant difference between smokers and controls in any site, with the exception of the buccal mucosa (Yu et al., 2017). Quitting smoking has been shown to increase bacterial diversity and alter community composition in both the mouth (Delima et al., 2010) and gut (Biedermann et al., 2013). Besides human cohort research, the gut microbiota has been shown to differ in tobacco smoke exposed mice, in comparison to air-only exposure (Wang, 2012; Allais et al., 2016).

The current study represents the first exploration of the effect of EC vapor and tobacco smoke exposure on the oral (buccal and saliva) and gut bacterial communities.

Materials and Methods

Study design and cohort

The study was approved by the Baylor College of Medicine Institutional Review Board (IRB H-38043). Written informed consent was obtained prior to collection of data and samples.

The cohort consisted of 30 individuals in three distinct exposure groups; EC users (n = 10), tobacco smokers (n = 10), and matched controls (n = 10). All participants were recruited from the Houston area. Inclusion criteria for EC users included daily use of ECs for at least six months. Inclusion criteria for tobacco smokers included an Fagerstrom test for nicotine dependence ≥4 and smoked a minimum of 10 cigarettes per day. Subject variables between the three exposure groups were comparable, with no significant difference in the sex, age, diet, height/weight, or race (Table 1). Notably, only 2/30 samples were from female participants. One EC user (EC7) reported occasionally smoking one tobacco cigarette per week and no other EC users reported use of tobacco cigarettes. EC7 had a comparable carbon monoxide (CO) ppm to other EC users and controls. No tobacco smokers reported use of EC. EC users vaped regularly throughout the day, used ECs daily, and had been actively using ECs for a median of three years.

Table 1 Subject information for the human cohort per exposure group.

	Controls	Electronic cigarette	Tobacco smoke	
Male sex	90%	90%	100%	
Age in years, median (IQR)	31 (28–36)	29 (24–37)	35 (30–45)	
Diet				
 Meat eater	90%	90%	100%	
 Vegetarian	10%	0	0	
 Vegan	0	10%	0	
Body mass index, median (IQR)	23.5 (22.5–24.5)	24.5 (22.5–26.7)	24 (21.5–25.5)	
Race				
 White	60%	70%	60%	
 Hispanic	10%	20%	10%	
 Asian	30%	10%	0	
 Black	0	0	30%	
Electronic cigarette				
 Nicotine concentration (mg), median (IQR)	–	9 (6–12)	–	
 Volume (ml)/day, median (IQR)	–	8 (3–19)	–	
 Years using, median (IQR)	–	3 (2–4)	–	
Tobacco smoke				
 Cigarettes/day, median (IQR)	0	0.2 (0.2–0.2)	14 (10–19)	
 FTND, median (IQR)	0	0	5 (4–6)	
 Carbon monoxide (ppm), median (IQR)	1 (1–2)	3 (3–4)	19 (14–24)	
Note:

IQR, interquartile range; FTND, Fagerstrom test for nicotine dependence.

DNA extraction

DNA was extracted from 125 mg of fresh fecal samples using the AllPrep Bacterial kit (Mo Bio 47054; Mo Bio Laboratories, Carlsbad, CA, USA) as per the manufacturers’ protocol. Entire buccal swabs and 500 μl saliva samples were extracted using the PowerMicrobiome RNA isolation kit (Mo Bio 26000-50; Mo Bio Laboratories, Carlsbad, CA, USA) as per the manufacturers’ protocol, omitting the necessary steps for co-elution of DNA and RNA, and with elution of nucleic acids in 50 μl.

16S rRNA gene sequencing

The bacterial 16S rRNA gene V4 region was amplified by PCR using barcoded Illumina adapter-containing primers 515F and 806R (Caporaso et al., 2012) and sequenced with the 2 × 250 bp cartridges in the MiSeq platform (Illumina, San Diego, USA). The read pairs were demultiplexed and reads were merged using USEARCH v7.0.1090 (Edgar, 2010). Merging allowed zero mismatches and a minimum overlap of 50 bases, and merged reads were trimmed at the first base with a Q ≤ 5. A quality filter was applied to the resulting merged reads and those containing above 0.5% expected errors were discarded. Sequences were stepwise clustered into operational taxonomic units (OTUs) at a similarity cutoff value of 97% using the UPARSE algorithm (Edgar, 2013). Chimeras were removed using USEARCH v7.0.1090 and UCHIME. To determine taxonomies, OTUs were mapped to a version of the SILVA database (Quast et al., 2013) containing only the 16S V4 region using USEARCH v7.0.1090. Abundances were recovered by mapping the merged reads to the UPARSE OTUs. A rarefied OTU table was constructed from the output files generated in the previous two steps for downstream analyses of alpha diversity, beta diversity (including UniFrac), and phylogenetic trends (Lozupone & Knight, 2005).

Statistical analysis

Samples were rarefied to 4,000 reads and rarefaction resulted in the loss of all negative controls for each DNA extraction kit. Analysis and visualization of bacterial communities was conducted in R (R Development Core Team, 2014). For analysis of alpha diversity and taxonomic relative abundance, the Kruskal–Wallis test (Kruskal & Wallis, 1952) was first applied to determine the overall statistical significance of the three groups. Only if the Kruskal–Wallis test showed a P < 0.05, pairwise significance was determined based on the Mann–Whitney test (Mann & Whitney, 1947). Differences in beta diversity (weighted Unifrac distance) were assessed using PERMANOVA. Linear regression was performed in R using the lm() function. When comparing more then one measure, such as multiple measures of alpha diversity or for multiple taxonomic genera, P-values were adjusted for multiple comparisons with the false discovery rate (FDR) algorithm (Benjamini & Hochberg, 1995).

All data and metadata files, as well as the R code used in the analysis, are provided in the Supplemental Information.

Results

Microbiota specific to sample site

Feces had a distinct bacterial profile compared to the oral samples (buccal swab and saliva) (Fig. S1A). The Shannon diversity indices showed a significant increase in saliva (P < 0.001) and feces (P < 0.001) in comparison to buccal swab samples (Fig. S1B). Dominant bacterial genera were also significantly different (P < 0.001) between the three sample types (Fig. S1C). Thus, analyses exploring the effects of tobacco smoking or EC use were stratified by sample type.

Alpha diversity of feces is reduced in tobacco smokers

The Shannon diversity was significantly reduced in fecal samples collected from tobacco smokers compared to controls (P = 0.009), but the number of observed OTUs was comparable between all groups (Fig. 1A). No significant difference was found in the number of OTUs or Shannon diversity between the groups in buccal swabs and saliva samples (Figs. 1B and 1C).

Figure 1 Boxplots of bacterial alpha diversity.

Analysis stratified per sample type. Controls (Con; orange); electronic cigarette (EC; blue); tobacco smoke (TS; green). Significance based on non-parametric Mann–Whitney test with FDR adjustment for multiple comparisons. Number of operational taxonomic units (OTUs) (A) and Shannon diversity (B) in feces. Number of OTUs (C) and Shannon diversity (D) in buccal swabs. Number of OTUs (E) and Shannon diversity (F) in saliva.

Bacterial profiles of feces and oral sites are significantly altered in tobacco smokers

Weighted UniFrac PCoA, a quantitative distance metric incorporating phylogenetic distances between taxa, showed tobacco smokers had significantly altered fecal bacterial profiles compared to controls (P = 0.027) and EC users (P = 0.009), but controls and EC users were not significantly different (P = 0.261) (Fig. 2A). This was consistent in buccal swabs, where bacterial profiles were significantly different between tobacco smokers compared to controls (P = 0.049) and EC users (P = 0.033), but controls and EC users were comparable (P = 0.886) (Fig. 2B). In saliva samples, the microbiota profiles of tobacco smokers and controls were significantly different (P = 0.046) and EC users were comparable to both tobacco smokers and controls (Fig. 2C).

Figure 2 Weighted UniFrac principal coordinate analysis (PCoA).

Analysis stratified per sample type. Controls (Con; orange); electronic cigarette (EC; blue); tobacco smoke (TS; green). Significance based on PERMANOVA. (A) Feces. (B) Buccal swab. (C) Saliva.

The relative abundance of bacterial genera was significantly associated with tobacco smoking in feces only

Fecal samples had a total of two genera significantly different between the three groups, with increased Prevotella (P = 0.006) and decreased Bacteroides (P = 0.036) in tobacco smokers (Fig. 3). Further pairwise comparisons of these genera showed Prevotella had significantly increased relative abundance in tobacco smokers compared to controls (P = 0.008) and EC users (P = 0.003), but no difference between EC users and controls (P = 0.99). Whereas Bacteroides showed significantly decreased relative abundance in tobacco smokers compared to controls (P = 0.017) and EC users (P = 0.003), but no difference between EC users and controls (P = 0.684). No significant difference in any bacterial genera was observed between the different groups in saliva or buccal swab samples (Fig. S2). These findings were also supported by correlations with CO levels, which reflect the amount an individual smoked tobacco cigarettes. Specifically, no genus was significantly associated with saliva or buccal swab samples, but Bacteroides was negatively correlated with CO level (P = 0.042) and Prevotella was positively correlated with CO levels (P = 0.011) (Table S1; Fig. S3).

Figure 3 Boxplot analysis of the bacterial genera in feces per exposure group.

Genera ordered based on lowest P value. All genera with >1% mean abundance included. Boxes represent interquartile ranges, with lines denoting median. Controls (Con; orange); electronic cigarette (EC; blue); tobacco smoke (TS; green). Kruskal–Wallis test with FDR adjustment for multiple comparisons showed two taxa significantly altered in feces, Prevotella (P = 0.006) and Bacteroides (P = 0.036). Mann–Whitney pairwise comparisons for Prevotella showed significantly increased relative abundance in TS compared to Con (P = 0.008) and EC (P = 0.003), but no difference between EC and Con (P = 0.99). Mann–Whitney Pairwise comparisons for Bacteroides showed significantly decreased relative abundance in TS compared to Con (P = 0.017) and EC (P = 0.003), but no difference between EC and Con (P = 0.684).

Discussion

This pilot study aimed to characterize the effects of EC vapor and tobacco smoke exposure on the bacterial profiles at multiple distinct and relevant body sites in a human cohort. To our knowledge this work represents the first study to concurrently explore the effect of EC vapor and tobacco smoke exposure on the microbiota. With users of ECs increasing at an unprecedented rate, it is imperative to understand the potential influences on host well-being, for which the oral and gut microbiota may have important consequences.

We report, for the first time, that regular use of ECs does not measurably influence oral or gut bacterial communities. However, compared to controls, tobacco smoking had a significant effect on the bacterial profiles in all samples analyzed, with the most significant associations found in the gut. This is in accordance with existing data showing the gut microbiota changes following smoking cessation (Biedermann et al., 2013, 2014). This is reflected in the alpha diversity analyses, where the fecal microbiota of tobacco smokers had significantly reduced Shannon diversity compared to controls. Previous studies have also showed the Shannon diversity is reduced in tobacco smokers compared to matched non-smokers in the gut (Opstelten et al., 2016), but recovers upon smoking cessation (Biedermann et al., 2013). Although smoking was recently reported to reduce buccal diversity (Yu et al., 2017), we found no difference in the diversity between the groups in buccal swabs. Overall, such studies provide further evidence for a direct effect of tobacco smoke in restricting microbial diversity and/or providing favorable conditions for specific taxa. The reduced bacterial diversity in the gut was striking, which may have important consequences for health and the risk of certain diseases. While inconclusive, reduced bacterial diversity has been associated with a range of conditions, including inflammatory bowel disease (Ott & Schreiber, 2006; Durbán et al., 2012; Sha et al., 2013), obesity (Turnbaugh et al., 2009), colorectal cancer (Ahn et al., 2013), and asthma (Abrahamsson et al., 2014).

Only the fecal microbiota was found to have specific genera significantly altered by exposure, with increased relative abundance of Prevotella, in accordance with existing data (Benjamin et al., 2012). Conversely, smoking tobacco cigarettes significantly decreased the relative abundance of Bacteroides compared to EC users and controls. Prevotella and Bacteroides are dominant members of the human gut microbiome (Arumugam et al., 2011; Koren et al., 2013; Gorvitovskaia, Holmes & Huse, 2016). Prevotella is associated with a high fiber diet and living in rural conditions (De Filippo et al., 2010; Ou et al., 2013; Tyakht et al., 2013; Kovatcheva-Datchary et al., 2015), whereas high Bacteroides abundance in the gut is generally attributed to a protein, fat, and sugar rich diet and a Western lifestyle (De Filippo et al., 2010; Ou et al., 2013). Prevotella and Bacteroides may have important implications for health and disease, with several species of the Bacteroides genus considered beneficial or probiotic (Xu & Gordon, 2003; Backhed et al., 2005). Existing evidence suggests intestinal inflammation, such as in Crohn’s disease, is associated with reduced abundance of Bacteroides (Guinane & Cotter, 2013). Furthermore, a reduced Bacteroides abundance has been associated with obesity in both humans (Ley et al., 2006) and mice (Ley et al., 2005; Turnbaugh et al., 2006) studies, but the direct role of the microbiome in obesity causality remains an area of active discussion (Sze & Schloss, 2016). Conversely, high Prevotella in the gut has been associated with human colon cancer (Chen et al., 2012; Sivaprakasam et al., 2016) and susceptibility to colitis (Elinav et al., 2011; Chow, Tang & Mazmanian, 2011).

No taxa were significantly altered in the oral (both buccal swab and saliva) microbiota. These results were surprising given the immediate proximity of the oral environment, relative to the gut, in smoke/vapor exposure. Indeed, smoking tobacco cigarettes has previously been shown to significantly alter the bacterial community in oral and lung samples (Charlson et al., 2010; Kozlowska et al., 2013; Mason et al., 2014; Wu et al., 2016). Conversely, existing studies have also reported no changes in smokers (Morris et al., 2013) and the taxa driving the separation vary between studies, which may reflect the differences in cohorts or methods, such as in specific site of sample collection, extraction, sequencing, and bioinformatics (Yu et al., 2017). Thus, further research in large multi-location cohorts is necessary to ascertain the direct effects of smoking across respiratory sites. Notably, both Prevotella and Bacteroides were highly specific to fecal samples (Table S2; Fig. S3), further demonstrating the precise effects of tobacco smoke exposure on taxa endogenous to the gut.

This study has several potential limitations. First, the cohort information was collected by questionnaire and while one EC user reported occasional use of tobacco cigarettes (one per week maximum), it is possible other participants used tobacco cigarettes and did not report this. However, to control for this we tested the CO levels (reflective of smoke inhalation) in all individuals and found tobacco smokers had higher CO ppm compared to EC users and controls, which would be expected (Table 1). Second, it is possible that the study was underpowered to detect subtle changes in the different sample sites and within some of the patient demographics. Third, only 2/30 participants in the study were female and, given the potential for sex-specific microbiota profiles (Haro et al., 2016), additional work is needed to determine if the findings differ between males and females. Further longitudinal work with frequent sampling in larger human cohorts is needed to validate the associations reported in this study and determine the potential mechanism and impact on host health. Despite an absence of taxonomic change in EC vapor exposure, determining potential changes to microbial and host functioning also represents an important area for subsequent research.

Conclusion

In summary, we found that tobacco smoking significantly alters the bacterial profiles in feces, buccal, and saliva samples. Compared to controls, exposure to ECs had no effect on the oral or gut communities. Changes in the gut microbiota of tobacco smokers were associated with increased relative abundance of Prevotella and decreased relative abundance of Bacteroides. From a microbial ecology perspective, this study supports the perception that ECs represent a safer alternative to tobacco smoking. However, other end points besides the microbiota will be important to consider when determining the impact of ECs on human health and disease. At a time when EC use continues to rise, we highlight the need for greater understanding on the direct short and long-term impact of exposure to vapor on the microbiome composition and function.

Supplemental Information

Supplemental Information 1 Correlations between the top 10 most abundant genera per sample type and the CO levels based on linear regression.

Click here for additional data file.

Supplemental Information 2 Mean relative abundance of significant fecal taxa from the human cohort.

Con, control; EC, electronic cigarette user, TS, tobacco smoker.

Click here for additional data file.

Supplemental Information 3 Analysis of bacterial profiles based on sample type.

(A) Weighted UniFrac PCoA. (B) Alpha diversity. (C) Boxplot of most significant bacterial genera. All genera in the box plot were significantly different by Kruskal-Wallis with a P < 0.001.

Click here for additional data file.

Supplemental Information 4 Boxplot analysis of the bacterial genera in oral samples per exposure group.

Genera ordered based on P value by reverse numerical order. All genera with >1% mean abundance included. No genera were found to be significantly different by exposure in either (A) Buccal swab or (B) Saliva.

Click here for additional data file.

Supplemental Information 5 Correlations between Bacteroides and Prevotella and the CO levels in each sample type.

Correlations are based on linear regression.

Click here for additional data file.

Supplemental Information 6 Biom file.

Click here for additional data file.

Supplemental Information 7 Metadata and alpha diversity values.

Click here for additional data file.

Supplemental Information 8 R markdown code for analysis.

Click here for additional data file.

Supplemental Information 9 tre file.

Click here for additional data file.

Supplemental Information 10 Weighted UniFrac values.

Click here for additional data file.

We also wish to thank the participants involved in this work.

Additional Information and Declarations

Competing Interests

Author Contributions

Human Ethics

Data Availability

The authors declare that they have no competing interests.

Christopher J. Stewart conceived and designed the experiments, performed the experiments, analyzed the data, contributed reagents/materials/analysis tools, prepared figures and/or tables, authored or reviewed drafts of the paper, approved the final draft.

Thomas A. Auchtung performed the experiments, authored or reviewed drafts of the paper, approved the final draft.

Nadim J. Ajami performed the experiments, authored or reviewed drafts of the paper, approved the final draft.

Kenia Velasquez performed the experiments, authored or reviewed drafts of the paper, approved the final draft.

Daniel P. Smith analyzed the data, authored or reviewed drafts of the paper, approved the final draft.

Richard De La Garza II conceived and designed the experiments, contributed reagents/materials/analysis tools, authored or reviewed drafts of the paper, approved the final draft.

Ramiro Salas conceived and designed the experiments, contributed reagents/materials/analysis tools, authored or reviewed drafts of the paper, approved the final draft.

Joseph F. Petrosino conceived and designed the experiments, contributed reagents/materials/analysis tools, authored or reviewed drafts of the paper, approved the final draft.

The following information was supplied relating to ethical approvals (i.e., approving body and any reference numbers):

The study was approved by the Baylor College of Medicine Institutional Review Board.

The following information was supplied regarding data availability:

The sequencing data generated in this study are available in the European Nucleotide Archive under project accession number PRJNA413706.

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
