# Peer review of "Effects of tobacco smoke and electronic cigarette vapor exposure on the oral and gut microbiota in humans: a pilot study"

_PeerJ, doi:10.7717/peerj.4693_

## Round 0.1 · original submission · Major Revisions

· Academic Editor

Major Revisions

While both reviewers find your manuscript to be timely and relevant, numerous points were raised that will need to be addressed before we can decide on publication. Of most importance are the statistical analysis methods applied. As you can see, specific recommendations are proposed by the reviewers to strengthen the analysis and presentation. In addition, there is need for additional information and discussion of the study subjects. Please also ensure that the raw data and code are available as pointed out by the reviewers.

Reviewer 1 ·

Basic reporting

• The use of EC and TS throughout the manuscript is quite confusing. Please spell out the words throughout the manuscript to make it easier to read. Abbreviations are ok in the figures as long as they are defined.
• Lines 47-48: Redundant use of the words “including especially”. Please clarify.
• Line 59: Please change “…community” to “communities”
• Line 129: Please change “…have been” to “…had been”
• Please change “a Prevotella” to “Prevotella” thoughout the manuscript.
• The reason for doing the study is very clear and other studies that have done similar analyses on tobacco smokers are referenced and addressed in the manuscript.
• Cannot locate raw data based on the project accession number provided by authors. Please provide exact URL that links to the page where the sequencing data can be found or searched for.
• I commend the authors for generating visually pleasing figures and supplemental information.
• Please address comments below regarding figures:
• Figure 3 title: This states that these are the top 10 most significantly altered genera? But, these are not all significant. Please clarify or change title.
• Please label boxplots (Figure 3 and Figure S3) with saliva, buccal, or feces so it’s easy to tell which location the graphs correspond to without referencing the text.
• Supplemental figures are missing legends.
• Figure S1: The boxplot colors in panel C are not defined. Consider changing the asterisk color so it isn’t so close in color to one of the boxplot groups.
• Italicize genera in Table S1.

Experimental design

• The research falls within the Aims and Scope of PeerJ (both biological and health sciences research.).
• The research question is clearly well defined and the authors provide information as to why they chose to do this study. The new field of microbiome research needs data! I am happy to see that the authors present site-specific characterization of the microbiome prior to examining treatment differences.
• Methods are described thoroughly and match those present in other epidemiology and microbiome manuscripts. IRB protocol is referenced.
• Please address comments below regarding the Methods section:
• Line 79: Please include information about the skewed sex-ratio of your participants (majority of study participants were male). This should be discussed, as sex-specific differences in microbiota have been shown in the literature.
• Like 81: Please provide the form of informed consent (verbal or written) received. If verbal, please state “Verbal informed consent…”. If written, please provide an empty copy of the consent form as a Confidential Supplemental Information file. This is requested based on PeerJ’s requirements.
• Line 84: Please define FTND. This is the first time it is mentioned in the manuscript.
• Line 88: Please define CO.
• Lines 93 & 94: Please include Catalog #’s for these DNA extraction kits in parentheses for easy reference.
• Line 115: This sentence is a bit unclear. Is this referring to negative controls for each DNA extraction kit?
• Line 120: Please define FDR. Also, please clarify this paragraph. Kruskal-Wallis p-values do not need to be adjusted for multiple comparisons. However, post-hoc test p-values (e.g. Mann-whitney) do need to be corrected.

Validity of the findings

• Cannot locate raw data based on the project accession number provided by authors. Please provide exact URL that links to the page where the sequencing data can be found or searched for.
• Line 40: This last sentence is too specific to microbiota. EC use may impact many other endpoints. Consider rewriting the sentence to include that other endpoints other than just microbiota composition and function should be assessed. There could be other detrimental impacts that are not captured by studying microbiota. Please also update the related sentence in the conclusion.
• Line 136-138: This statistical analysis does not tell you if they are “significantly increased.” What can be said, however, is that there were statistical differences between the 3 groups. And that diversity was higher in x, vs. y. See general comment about doing post-hoc tests to see which groups are different.
• Figure 1: See general comments on statistical analysis. Kruskal-wallis test should be used first and then, if that p-value is significant, a post-hoc test with multiple comparison adjustments can be conducted. This approach may help simplify your boxplots, as you can report a single p-value for the K-W test on the graph, and then use different letters to indicate pairwise differences if they exist.
• Lines 160-161: These results would benefit from making pairwise comparisons to show that Prevotella and Bacteroides were significantly higher or lower in tobacco smokers feces.
• Figure S1A and Figure 2: Nonparametric statistical analyses should not be used for beta-diversity metrics. A statistical test such as PERMANOVA or ANOSIM should be conducted to assess the global differences between the 3 groups instead.
• Lines 162-164: For some races, diet groups, etc there was a n=0 or 1 person. Therefore, there are not enough individuals to make the claim that there are no statistical differences based on these endpoints. Please remove this statement or say it in another way.
• Line 182: What alpha diversity endpoint was measured in the referenced study?
• Lines 185-186: This “statistical difference” (e.g. increased OTUs in TS) may be eliminated if a Kruskal-Wallis test is done first to compare all 3 groups. Please see general comments about statistics.
• Lines 202-204: Same comment as above. Based on the number of individuals, this claim should either not be made, or be more carefully made.
• Line 204: Lim et al 2016 said sputum microbiota was influenced by tobacco smoke. In this study, you did not see any impacts on saliva. Please address why these differences are observed.

Additional comments

This is an original article that characterizes changes in microbial communities in different regions (oral and gut) in a pilot study of a small human cohort of controls, electronic cigarette users, and tobacco smokers. I am particularly elated that the authors characterized the overall differences in their sampling site communities prior to analyzing differences between treatment groups, as this is typically overlooked and provides a baseline understanding of the communities across sampling sites to help understand the inherent microbiome variability.

The overall study design is limited, but this is clearly a pilot study and the authors do a nice job of addressing some of these limitations in the discussion. But, the lack of mention regarding the majority of the subjects being males was not addressed, and is particularly important given that other studies have shown sex-dependent differences in baseline microbiota and responses to toxicants.

The main findings of this article are 1) no impacts of electronic cigarette smoking were found on the microbial communities in the oral cavity (buccal and saliva) or gut (feces) and 2) tobacco smokers show significant changes in overall microbial community composition and increases or decreases in certain bacterial taxa, particularly in the feces. However, I have major concerns about the statistical analyses used to analyze the data in the manuscript. Beta-diversity metrics (PCA plots) should be analyzed using a PERMANOVA or ANOSIM to assess the global differences between the 3 groups. Alpha diversity metrics and individual taxa box plots can be analyzed with a nonparametric test as was done here, but the rule of thumb is that for 3 groups, a Kruskal-Wallis test should be done first and if the p-value is significant, then pairwise comparisons (Dunn or Mann-whitney) can be made with multiple comparison adjustments. In some cases, doing the analysis this way may benefit the authors and make the results less confusing (e.g. increased OTUs in tobacco smokers).

Other Specific General Comments
• Please consider including a table of alpha diversity metrics for each group, and including richness and evenness.
• Lines 124-132: This section is almost completely redundant with the materials and methods study cohort paragraph. Please consider combining these paragraphs so all information in both is retained. It seems reasonable to put it in the methods section as defining the study cohort.
• Line 130: should “medium” be “median”? I also think this is an inaccurate measurement. Why did the authors choose to report the median and not the average and standard deviation instead?

·

Basic reporting

This study describes the effects of tobacco smoke and electronic cigarette vapour exposure in humans. This is an important study and it is well executed, although the presentation of the data and the statistical analysis can be improved. The writing style is clear.

The experimental design is very limited due to the low number of individuals included in this study. The results are promising but not likely to bring reliable conclusion. If this is really a pilot study, my understanding is that the authors will perform a bigger study. In this case, a publication as a preprint would be more suitable. However, this is only a recommendation. My role as a reviewer is not to comment on the publishing strategy.

I am not able to give a conclusion on whether or not I think that this study should be published because it needs to be extensively revised to be assessed at face value.

Bioinformatics pipelines for microbiome studies are relatively immature and several recent reviews emphasise the lack of consistency between the different pipelines. This is why the analysis should be fully reproducible. The authors should include their raw data (sequencing data posted on a data repository), their processed data (such as an OTU table), and the code used to process their data as a supplementary material.

Experimental design

The research question is very relevant and well addressed (for a pilot study). The statistics and the presentation of the data can nonetheless be improved. I believe that a minimal requirement would be that the authors present all the raw data, their processed data and provide their code in order to allow a full reproducibility of the study as stated above.

The microbiome is also influenced by alcohol consumption, and by the lifestyle. Where are these people living? Living in a polluted city could be a major confounding factor. Covariates should be more controlled. The authors report the measurement of CO levels. More details should be provided. It is very important to obtain a reliable biomarker of smoke ingestion since an EC user will be likely to go out with other individuals using conventional cigarette and thus be passively exposed. The authors should also run an association study to see if the CO levels correlate with relative abundance values in bacteria.

Validity of the findings

The authors should improve the presentation of their statistics. Is it not clear how the authors performed the adjustment for multiple comparisons. This is a general problem for microbiome studies since the data has a tree structure and it is thus difficult to understand how many independent tests are actually made.

Better statistical pipelines exist for pair-wise comparisons of gut microbiome compositions. A hierarchical multiple testing would be more suitable. Alternatively, the authors can adopt a multivariate strategy and use an sPLS-DA.

The presentation of the box plots could be improved by adding the actual data dots. It looks like the authors used R for the preparation of the figures. It is easy to add the data dots on the box plots.

The BMI should be presented; it is a better indication than the weight or the height of individuals. Is any of the covariate associated with relative abundance values?

More details on the overall structure of the microbiome can be provided. How many taxa were actually determined? The tree structure of the data is important to take into account. Thus, it would be important to present a tree of the different microbiome in order to understand the representation of the different taxonomic levels in this dataset. This can be a supplementary material.

---

## Round 0.2 · Minor Revisions

· Academic Editor

Minor Revisions

Thank you for your quick turnaround on the revision and for thoroughly addressing the reviewers concerns. Reviewer 1 points out some suggested small changes and corrections that should be addressed before final acceptance.

Reviewer 1 ·

Basic reporting

I am still not able to see supplemental figure legends. Please provide these with supplemental figures.

Figure 2 legend: “…reverse numerical order” is not clear.

Experimental design

Lines 90-93: Is this EC6 or 7 supposed to be the same EC#?

Lines 96-101 are still repetitive to lines 83-95 and can be condensed into that paragraph or removed.

Line 258: Thank you for addressing the male/female ratio. Consider replacing the words “are consistent in females” with “differ between males and females” to clarify what is meant here.

Line 136: Please add the acronym “FDR” after “…false discovery rate” since this acronym is used in subsequent figure legends.

Validity of the findings

Beta-diversity analysis: Please report overall PERMANOVA p-value for Figure 2. Were the pairwise p-values reported in Figure 2 adjusted for multiple comparisons (e.g. FDR)?

Lines 221-223: The following sentence needs to be removed from the discussion because the previous statement about significant effects of diet and lifestyle was removed due to the sample size being too low to make this claim: “Notably, diet and lifestyle was not found to drive the significance in this study, which is in accordance with Lim et al. (2016) who found that smoking was the most important lifestyle factor influencing the sputum microbiota (Lim et al., 2016)”

Additional comments

Thank you for addressing the issues raised previously and for providing the raw data. I commend the authors for their efforts. The CO data is also a great addition based on the other reviewer’s recommendation. I have a few other small recommendations that would, ideally, be addressed prior to publication:

Lines 93-95: The following sentence was moved from the results to methods, which is great. However, it still says “medium”. Should this be “median”?: “Electronic cigarette users… for a medium of 3 years”

Lines 176-178: “increased” should be “decreased” in the following: “Whereas Bacteroides showed significantly increased relative abundance in tobacco smokers compared to controls (P = 0.017) and electronic cigarette users (P = 0.003), but no difference between electronic cigarette users and controls (P = 0.684).” Please adjust in the figure legend too.

Line 266: Please correct spelling of “salvia”

Is the change in feces microbiota following tobacco smoke use some sort of stress response? The results presented here correspond with others in the literature, and are quite interesting.

·

Basic reporting

no comment

Experimental design

no further comment

Validity of the findings

no comment

Additional comments

The authors have taken into account all my comments. The manuscript is much better now. I am also glad to see that the authors have disclosed the totality of their raw data and their code. I wish every paper could be as transparent as this one.

---

## Round 0.3 · accepted · Accept

· Academic Editor

Accept

Thank you for your prompt turnaround of the revisions and to your attention to the suggestions from the reviewers.

#